# Double boron-embedded multiresonant thermally activated delayed fluorescent materials for organic light-emitting diodes

Kenkera Rayappa Naveen [1,2], Hye In Yang[1,2] & Jang Hyuk Kwon [1✉]

The subclass of multi resonant thermally activated delayed fluorescent emitters (**MR-TADF**) containing boron atoms has garnered significant attention in the field of organic light emitting diode (OLED) research. Among boron-based **MR-TADF** emitters, double boron-embedded **MR-TADF** (**DB-MR-TADF**) emitters show excellent electroluminescence performances with high photoluminescence quantum yields, narrow band emission, and beneficially small singlet-triplet energy levels in all the full-color gamut regions. This article reviews recent progress in **DB-MR-TADF** emitters, with particular attention to molecular design concepts, synthetic routes, optoelectronic properties, and OLED performance, giving future prospects for real-world applications.

Since 2012, thermally activated delayed fluorescence (TADF) emitters have received significant attention for their application in organic light-emitting diodes (OLEDs), replacing the expensive noble metal atom-based phosphorescent emitters[1–10]. These emitters usually exhibit a small singlet-triplet energy gap ($\Delta E_{ST}$), which facilitates spin flipping of the reverse intersystem crossing (RISC) in excited states[11,12]. Primarily, conventional TADF emitters are designed based on donor-acceptor (D-A) configuration[1,13–15]. Such combinations of donor and acceptor play a significant role in electron density distributions and allow the clear separation between the highest occupied molecular orbital (HOMO) and lowest unoccupied molecular orbital (LUMO), resulting in reduced $\Delta E_{ST}$ values[2,3,16–19]. As of now, many TADF emitters are showing tremendous performances in all the color regions[16,20,21]. However, most of the TADF emitters are showing intramolecular charge transfer (ICT) characteristics, which cause structural relaxation in the excited state, and large stokes shift, and broad emission spectra[6,22–29]. Such broad emissions are detrimental to achieve high color purity for future OLED applications[3,7,30].

Usually, the CIE (International Commission on Illumination) coordinates envisaged the full-color regions that can be obtained by combining the three primary colors (red (R), green (G), and blue (B)) by varying the wavelength and emission intensity[31]. However, recently, the International Telecommunication Union (ITU) announced a new color gamut standard for ultra-high-definition TV (UHDTV) called the Broadcast Service Television 2020 (BT 2020) with CIE coordinates for the R, G, and B colors of (0.708, 0.292), (0.170, 0.797), and (0.131, 0.046), respectively[32,33]. So to achieve such deep-level color coordinates, the emitters required extremely narrowband emission spectra. Among the reported TADF emitters, the boron-based D-A configured push-pull materials are efficiently performed in OLEDs, but rather they also exhibit FWHM ~50 nm[34]. Such detailed boron-based materials were already summarized in our previous review papers[34,35]. In 2016, Hatakeyama et al. unveiled a new molecular design concept based on the multiple resonance effect of boron (B) and nitrogen (N) atoms and developed **DABNA-1**[36]. This design allows the HOMO and LUMO separation with opposite resonance effects on the same π-conjugated plane. Usually, this multi-resonant TADF (**MR-TADF**)

[1] Organic Optoelectronic Device Lab (OODL), Department of Information Display, Kyung Hee University, 26, Kyungheedae-ro, Dongdaemun-gu, Seoul 02447, Republic of Korea. [2] These authors contributed equally: Kenkera Rayappa Naveen, Hye In Yang. ✉email: jhkwon@khu.ac.kr

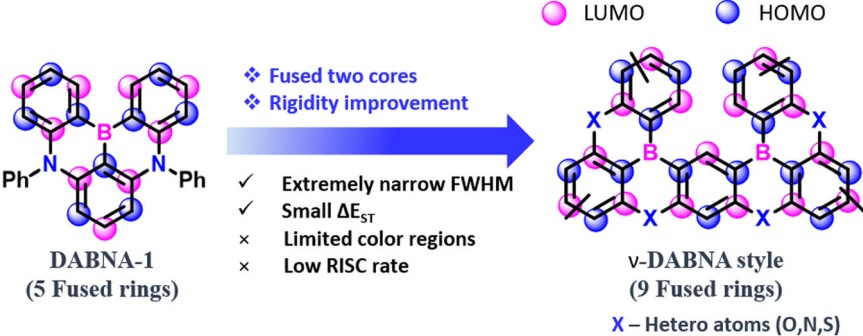

**Fig. 1 DB-MR-TADF vs SB-MR-TADF.** Merits of double boron embedded MR-TADF (DB-MR-TADF) emitters compare to single boron MR-TADF (SB-MR-TADF) materials.

manifested short-range CT (SRCT) character from their LUMOs localized on the electron-withdrawing atoms (B) and at the ortho and para positions relative to them. At the same time, HOMOs will be restricted on the electron-donating atoms (N or O) and at the meta positions relative to electron-withdrawing atoms[37,38]. This MR effect-suppressed vibronic coupling leads to a narrow full width at half maximum (FWHM), and the rigid molecular structure helps realize a minimal Stokes shift below 30 nm, resulting in a low reorganization energy and high photoluminescence quantum yield (PLQY)[39–44]. Notably, compared to conventional TADF emitters, the **MR-TADF** emitters have significant advantages such as high color purity (high PLQY) and narrow emission spectra with FWHM of ~30 nm[35,38,39,45–47]. As of now, many groups developed over 100 molecules in **MR-TADF** emitters. Among the reported boron-embedded **MR-TADF** compounds, the double boron-embedded **MR-TADF** (**DB-MR-TADF**) emitters are performing tremendously in OLEDs compared to single boron-embedded **MR-TADF (SB-MR-TADF)** materials[33,48–55] (Fig. 1). For example, in the blue region, the emitter **v-DABNA** shows extremely narrow FWHM and high external quantum efficiency (EQE) of ~35%[56]. Even in green (**v-DABNA-CN-Me**)[57] and red (**BNO1**)[58] color regions, the maximum EQE is crossed over 30% in the benchmark of **DB-MR-TADF** emitters.

In this review, we mainly focus on the recent developments in **DB-MR-TADF** emitters based on full-color regions. We categorize these materials according to their respective colors (RGB). Rational molecular design strategies to achieve narrow PL and EL emission and the related electronic structure and optoelectronic characteristics, including the resulting device performances, are discussed to understand the underlying mechanisms for controlling the emission bandwidth. Further, we even discussed the synthetic aspects utilized for **DB-MR-TADF** emitters. Finally, we provide our summary and future perspectives on the remaining challenges in this research area that must be overcome to develop the next generation of wide-color gamut OLED displays.

## Blue DB-MR-TADF emitters
Firstly, Hatakeyama et al. developed inverted DABNA-type emitters, which are azadiboranaphthoanthracenes (**ADBNAs**) based on two boron and one nitrogen atom; which are comparable to conventional DABNA analogs[59]. The synthesis was succeeded by lithiated with tert-butyllithium followed by boron tribromide in the tert-butylbenzene solvent. Further, the treatment of magnesium bromide type precursor can yield both emitters, namely, **ADBNA-Me-Mes** and **ADBNA-Me-Tip**. Both the emitters exhibited PLmax of ~480 nm in DOBNA-OAr film. Importantly these materials showed red-shifted emission compared to **DABNA-1** (PLmax: 460 nm). Furthermore, both

materials manifested a small $\Delta E_{ST}$ of ~0.20 eV, indicating the presence of better TADF. When the OLED device is fabricated with both materials with the support of DOBNA-OAr (as host), the EL manifested sky blue emission with ELmax of ~480 nm with FWHM of less than 33 nm. The ADBNA-Me-Tip showed a better EQEmax of 21.4% compared to ADBNA-Me-Mes (EQEmax of 16.2%) due to an increase in light outcoupling efficiency and molecular orientation factors. It is also observable that both materials' approach is better compared to DABNA-1 as they exhibit bathochromic shift with better EL performances and low roll-off characteristics.

Later, Wang et al. reported a series of ADBNA materials[60] by incorporating a donor (nitrogen) center in a triangulene shape, similar to the ADBNA series reported by Hatakeyama et al. (Fig. 2). The synthesis of all the designed materials had symmetric and unsymmetrical nature. Among all materials, **6b** showed significantly less $\Delta E_{ST}$ of 0.13 eV in the 1 wt% PMMA matrix; however, a low PLQY of 18% was observed. Compounds 3–5b showed **MR-TADF** characteristics with positive solvatochromism and narrow FWHM. In the case of 6a and 6b, there is an observable insight that two emission peaks were observed. The emissions were located at 477/481 and 609/601 nm in the dichloromethane solution. In the case of 3, the blue emission band is like **MR-TADF** in nature with a narrow emission spectrum. But in the case of 6b, it showed a low energy emission band showing strong ICT characteristics like conventional D-A type emitters. In this work, authors mainly focused only synthesis point of view; no devices were fabricated with the designed emitters to identify the EL performances. However, from the overall point of view, these ADBNA-type emitters are one type of class with an alternative in the finding **MR-TADF**. However, from a device point of view, these are still not so good and sound for real applications.

Hatakeyama et al. developed new emitter's through multiple borylation methods. Among them, the **B2** shows a higher synthetic yield of 80% and exhibited sky blue emission with the PLmax of 455 nm and the FWHM of 32 nm[61]. Further, the nature of the material is analyzed by X-ray crystallography, and it manifests helical nature. The OLED is fabricated with a mCBP host, and B2 as emitter exhibited a high EQEmax of 18.3% with the ELmax of 460 nm and CIE coordinates of (0.13, 0.11). From this borylation method, it is understood that double or triple boron incorporation is possible by using BBr$_3$ or BI$_3$ with the support of a Hünig base in reducing solvents. Even though this design allows for new insights into the synthetic perspective, it is tough to proceed for large-scale purposes.

Again in 2019, the same group reported a fully fused system consisting of five benzene rings connected by two boron and four nitrogen atoms and two diphenylamine groups named it N7,N7,N13,N13,5,9,11,15-octaphenyl-5,9,11,15-tetrahydro-5,9,11,15-

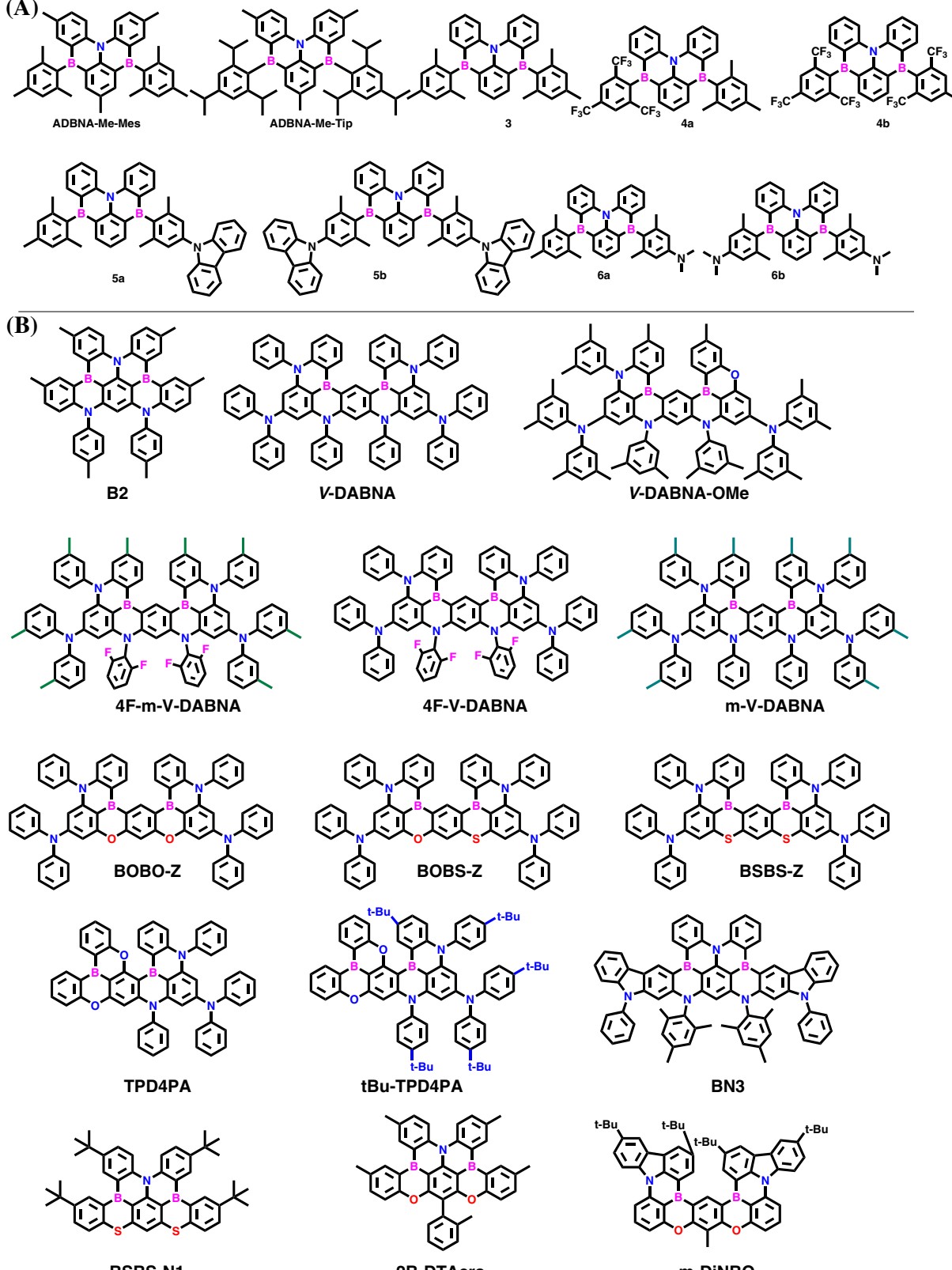

**Fig. 2 Chemical structures of ADBNA & ν-DABNA style.** Molecular structures of the (**A**) azadiboranaphthoanthracenes (ADBNAs) and (**B**) N7,N7,N13,N13,5,9,11,15-octaphenyl-5,9,11,15-tetrahydro-5,9,11,15-tetraaza-19b,20b-diboradinaphtho[3,2,1-de:1′,2′,3′-jk]pentacene-7,13-diamine (**ν**-DABNA) style based Blue **DB-MR-TADF** emitters.

tetraaza-19b,20b-diboradinaphtho[3,2,1-de:1′,2′,3′-jk]pentacene-7,13-diamine (**υ-DABNA**), which has been benchmarked in **MR-TADF** emitters till date[56]. In this design, the boron is surrounded by nitrogen atoms with proper positions to observe the clear MR effect in alternative atoms. The MR effect is regarded as expected on boron/nitrogen atoms, which induce the HOMO and LUMO density localizations on different atoms. This even minimizes their bonding/antibonding character and vibrational coupling between ground and excited states. Such a feature allows getting the extremely narrow FWHM spectra. The synthesis of **υ-DABNA** is done in three steps by using simple available commercially starting compounds with the support of palladium-catalyzed reactions. The foremost critical final step involves the one-pot double borylation using the tandem bora Friedel crafts reaction (BBr₃). As a result, the **υ-DABNA** manifested blue emission with the PL$_{max}$ of 469 nm and FWHM of 14 nm in toluene solution. Furthermore, this design indicates that joining the fused **DABNA-1** analogs[36] increases the fused aromatic ring systems as well. Additionally, this design gives an extremely small $\Delta E_{ST}$ value of 17 meV, which is the best in **MR-TADF** emitters. **υ-DABNA** also manifested a high RISC rate constant of $2.0 \times 10^5$ s⁻¹, which is higher than conventional previous report DABNA analogs. Furthermore, **υ-DABNA** also showed a high PLQY of ~90% with highly bright blue emission. The OLED fabricated with 1 wt% **υ-DABNA** and 99 wt% DOBNA-OAr (as a host) showed tremendous performances. The EL characteristics showed sky blue emission with EL$_{max}$ of 469 nm with narrow FWHM of 18 nm. Furthermore, the OLED exhibited a maximum EQE of 34.4% with CIE coordinates of (0.12, 0.11). This work shows excellent EL performances compared to other reported materials in blue color. From the design point of view, it is clearly understood that this type of design allows getting the high PLQY, extremely narrow FWHM, and high RISC values. However, In the case of **υ-DABNA**, the CIE y coordinate is 0.11, which deviates from NTSE and BT2020 pure blue color requirements.

Later same group developed **υ-DABNA-OMe** to shift the CIE y coordinate towards pure blue color[62]. The design is based on the utilization of an oxygen atom instead of one nitrogen atom in the **υ-DABNA** core. Due to the lower atomic energy, the oxygen atoms deviate the HOMO energy compared to the LUMO energy, which results in the band gap increment. The synthesis of the **υ-DABNA-OMe** succeeded with the support of BI₃ instead of BBr₃. The **υ-DABNA-OMe** exhibited sharp pure blue emission at 464 nm with the FWHM of 19 nm in toluene solution. Precedently, the **υ-DABNA-OMe** showed a 5 nm blue-shifted compared to υ-DABNA. The $\Delta E_{ST}$ value of 29 meV is higher compared to the **υ-DABNA** $\Delta E_{ST}$ value; however, blue-shifted emission is obtained. The OLEDs were fabricated with the help of **DOBNA-Tol** as host and the **υ-DABNA-OMe** as dopant (with 1 wt%). The device showed a high EQE of 29.5% with a pure blue emission at 464 nm and FWHM of 24 nm. Notably, **υ-DABNA-OMe** displayed alleviated efficiency roll-off of only 2.6% at 1000 cd/m² compared to **υ-DABNA** (8.4%) due to the reduced bimolecular quenching process like TTA/TPA. Consequently, the **υ-DABNA-OMe** manifested a CIE y coordinate of 0.10, which is very near to NTSE pure blue requirements. Notably, the **υ-DABNA-OMe** revealed a long lifetime (LT$_{50}$) of 314 h at 100cdm⁻² compared to the previous **υ-DABNA** (31 h). This indicates that the **υ-DABNA-OMe** is ten times longer in lifetime concern compared to **υ-DABNA**. Even though this **υ-DABNA-OMe** showed blue-shifted emission compared to **υ-DABNA**, there are still some disadvantages. Such as, FWHM increased from 14 to 19 nm, $\Delta E_{ST}$ risen from 17 to 29 meV, and RISC decreased from $2.0 \times 10^5$ to $1.6 \times 10^5$ s⁻¹. Furthermore, CIE y of 0.10 is still insufficient for NTSE and BT 2020 pure blue color requirements.

To improve the color purity, later on, Yasuda et al. unveiled the new molecular design by replacing the intracyclic nitrogen with oxygen/sulfur atoms. They developed by utilizing two oxygen atoms (**BOBO-Z**), one oxygen and one sulfur (**BOBS-Z**), and two sulfur atoms (**BSBS-Z**)[63]. All the designed emitters are expected to show the blue-shifted emission as they contain the weak electron donating ability atoms (O or S). All three emitters were synthesized via double tandem lithiation–borylation–annulation reactions. Consequently, all three emitters exhibited PL$_{max}$ ranges from 441 to 460 nm with narrow FWHM. All the emitters showed blue-shifted emissions compared to υ-DABNA and **υ-DABNA-OMe**. Notably, BOBO-Z, BOBS-Z, and BSBS-Z demonstrated intense ultrapure blue emissions with CIE color coordinates of (0.15, 0.03), (0.15, 0.06), and (0.14, 0.07), respectively. Such low CIE y coordinates match near NTSE and BT2020 pure blue color requirements. Among the three, the sulfur-doped materials **BOBS-Z** and **BSBS-Z** show improved PLQY compared to conventional **υ-DABNA**. Due to the heavy atom (sulfur), the BOBS-Z and BSBS-Z manifested significantly high RISC rate values of $8.6 \times 10^5$ and $1.6 \times 10^6$ s⁻¹, respectively, which were 2.6 and 4.9 times higher than that of **υ-DABNA**. Further, with the support of the **mCBP** host, the OLEDs were fabricated. As like in the PL emission, the EL of the compounds showed pure-blue emission, 445 (BOBO-Z), 456 (BOBS-Z), and 463 nm (BSBS-Z), and narrow FWHM falls in the range of 18–23 nm, which indicates expected hypsochromic shift compared to the **υ-DABNA** (472 nm). The intriguingly, OLEDs showed excellent color purity with CIE(x, y) of (0.15, 0.04) for BOBO-Z, (0.14, 0.06) for BOBS-Z, and (0.13, 0.08) for BSBS-Z fulfilling the requirements of Rec.2020 standard of UHD OLEDs. The S-containing derivatives showed stand-out performance with EQE$_{max}$ of 26.9 and 26.8% over BOBO-Z (13.6%) and **υ-DABNA** (26.6%) due to their fast RISC mechanism. Initially, the device stability was analyzed in the unipolar host, **mCBP**, and showed a very short lifetime; further, it was improved (LT$_{50}$ > 30 h at 100 cd/m²) by replacing the **mCBP** with bipolar host **mCBP-CN**. Though authors achieved considerable blue shift and excellent color purity compared to **υ-DABNA** with the key to design heteroatom interplay, they exhibited low EQE ≤ 27%, and especially BOBO-Z is showing very less EQE of less than 14% in the deep blue color region.

Naveen et al. unveiled a new molecular design by incorporating the fluorine atoms and methyl groups in conventional **υ-DABNA** and designed three emitters, namely, **m-υ-DABNA**, **4F-υ-DABNA**, and **4F-m-υ-DABNA**, respectively[64]. Using methyl groups at *para* positions to the boron atoms and fluorine atoms at *ortho* positions to the intracyclic nitrogen (N) atoms resulted in bandgap enhancement by electron donating and electron withdrawing effects. These materials were synthesized using a stoichiometric amount of BBr₃ and *o-Dichlorobenzene* solvent at a high temperature (190 °C) and succeeded in one-pot double boron-cyclizations. Comparatively, all three emitters showed a high synthetic yield of over 32%, indicating ease for commercial scales ups. Furthermore, all these three emitters exhibited high decomposition temperatures ($T_d$) over 480 °C, indicating their molecular stability for real display applications. Further, as expected, the PL spectra in toluene solution displayed strong and narrow blue emission maximum peaks at 464, 457, and 455 nm for **m-υ-DABNA**, **4F-υ-DABNA**, and **4F-m-υ-DABNA**, respectively. Notably, all these areas manifested hypsochromic shifted emission with the extremely narrow FWHM of 14 nm. All three emitters even showed extremely low $\Delta E_{ST}$ values of less than 0.07 eV, indicating their ability to TADF process. The PLQY of all these materials is maintained to be high at ~90%. Interestingly the RISC rate was improved for all three emitters compared to conventional **υ-DABNA**. Further, the OLEDs were fabricated by using **DBFPO** as a host due to the advantage of high triplet

energy. All three emitters showed extremely narrow blue emissions with an FWHM of only 18 nm, indicating their high molecular rigidity. Consequently, all three emitters exhibited CIE coordinates of (0.12, 0.12), (0.13, 0.08), and (0.13, 0.06) for **m-v-DABNA**, **4F-v-DABNA**, and **4F-m-v-DABNA**, respectively, approaching the clear NTSE and BT2020 pure blue color requirements. Furthermore, all three emitters revealed a high EQE of over 33% in pure blue color, which is far better than **BOBO-Z** moiety[63]. These results are benchmarked in pure blue **MR-TADF** OLEDs with the CIE ≤ 0.1 with a narrow FWHM of 18 nm.

Even though **v-DABNA** modified analogs showed the best performances, some drawbacks still exist, such as high molecular weight and low RISC and roll-off characteristics. To overcome this, Naveen et al. recently developed new blue emitters, namely **TPD4PA** and **tBu-TPD4PA**[65]. The core is somewhat new considering the highly stable **DOBNA** moiety in the proper molecular design[34,66–69]. The **TPD4PA** is designed by amalgamating the high CT characteristic **DOBNA** and MR-type emitter **PAB**[70]. This design restricts the HOMO energy due to rigid oxygen **DOBNA** and increases the bandgap, witnessing the pure blue emission. The fully fused TPD4PA/tBu-TPD4PA was synthesized using commercially available starting materials in four steps. Among them, the final one-pot double borylation was done using the $BBr_3$ and *1,2,4-trichlorobenzene* solvent at a high temperature (180 °C). Interestingly, unlike other **MR-TADF** emitters, both emitters have resulted in a high synthetic yield of ~25%. As expected, the optical bandgap (2.75 and 2.71 eV for TPD4PA and tBu-TPD4PA) was increased compared to v-DABNA (optical bandgap: 2.60 eV). Compared to other reported double boron **MR-TADF** emitters, these materials showed tremendous improvement in thermal stability as they revealed high decomposition temperatures ($T_d$) of 521 °C for TPD4PA and 515 °C for tBu-TPD4PA, respectively. Both the emitters showed the $PL_{max}$ of 445 (TPD4PA) and 451 nm (tBu-TPD4PA) in toluene solution with the FWHM of 19 nm for both. Comparatively, the small $\Delta E_{ST}$ values are also retained with this design concept. Mainly, both the emitters exhibited a high RISC rate ~$2.51 \times 10^5\,s^{-1}$, which is higher than single boron **MR-TADF** (DOBNA, DABNA-1, and PAB) and double boron **MR-TADF** emitters (v-DABNA, v-DABNA-OMe, and BOBO-Z) also. A further interesting fact is that the TADF mechanism occurred via three steps, including (i) IC from $T_1$ to $T_2$, (ii) RISC from $T_2$ to $S_1$, and (iii) fluorescence from $S_1$. This was clarified with the support of SOC calculation, indicating the $\langle S_1|\hat{H}SOC|T_2\rangle$ is almost 3.3 times of $\langle S_1|\hat{H}SOC|T_1\rangle$ value for both materials. Further, the TADF-OLEDs were fabricated using **mCBP-CN** as a host due to its ET-type character with deep LUMO energy level. The OLEDs showed maximum EQEs of 30.7 and 32.5% for **TPD4PA** and **tBu-TPD4PA**. Both the devices exhibited narrow band deep blue emission with the CIE y coordinates match near NTSC and BT2020 blue color requirements.

From the above discussion, it is understood that all the designs only restricted the HOMO/LUMO at specific positions of the conventional **v-DABNA**. So based on that, here we gathered all the design concepts based on **v-DABNA** and depicted in Fig. 3. The advantages of utilizing **v-DABNA** are highly recommendable for real commercial applications. However, the other design concepts can also pave the pathways for future development design with good new ideas.

Meanwhile, Chuluo yang et al. recently developed three deep blue **MR-TADF** emitters (**BN1-BN3**) featuring fully fused structures[71]. In this, they achieved all the targets from the same precursor. Due to the molecular rigidity of all three emitters, more negligible structural relaxation and narrower spectra can be expected. Among all the materials, **BN1** showed a high synthetic

yield of 68% compared to other emitters, **BN2** (39%) and **BN3** (55%). This is due to the HOMO delocalization major at the carbon atoms of the central benzene position. In the case of **BN3**, the utilization of 24 equivalents of $BBr_3$ with a high temperature only occurs. Interestingly, **BN3** adopted a more twisted scaffold due to unique B, N, B-doped [4] helicene. Notably, all the emitters exhibited pure and narrowband deep blue emissions with an FWHM of less than 18 nm in toluene solution. Furthermore, all three showed significantly lower $\Delta E_{ST}$ values of ≤0.20, indicating the effectiveness of the TADF mechanism. The **BN3** shows a very high PLQY of 98% compared to all the reported double boron-embedded emitters. This is due to the increased molecular rigidity with the double helicene structures. This also helped to grow the RISC constant values. This material (**BN3**) also confirmed that the RISC mechanism might occur via the $T_2$ state as the SOC values of $S_1$ and $T_2$ are higher than those of $S_1$ and $T_1$. The OLEDs were fabricated by adopting the hyperfluorescence system. The emissive layer contains **DBFPO**:25 wt% **3Cz2BN**:1 wt% of **BN1-BN3**. The OLEDs manifested narrowband pure blue emission at 457, 467, and 458 nm with FWHM values of 28, 23, and 23 nm for **BN1**, **BN2**, and **BN3**, respectively. Compared to **BN1** and **BN2**, **BN3** showed tremendous performance with the $EQE_{max}$ of 37.6% due to the improved RISC process. Furthermore, the CIE y coordinate of **BN3** is led 0.08, which also matches NTSE pure blue color requirements. So from this contribution, we can understand that enhancing the molecular rigidity by extended skeleton would benefit smaller reorganization energy to narrow the emission spectrum.

Yasuda et al. reported a new molecular design based on nano graphitic fused nonacyclic π-system (**BSBS-N1**) as another double-boron embedded material class. The structure contains two boron and sulfur atoms with one nitrogen atom in a fused system[72]. The **BSBS-N1** was synthesized in five steps from the commercially available starting materials. The critical step, cyclization, is done using the one pot double tandem lithiation and borylation annulation, which also resulted in a reasonable yield of 18%. Notably, the **BSBS-N1** displayed pure sky blue emission at 478 nm with an FWHM of 21 nm and a high PLQY of 89%. Even though the heavy atoms are incorporated, the $\Delta E_{ST}$ value is meager as 0.14 eV, indicating their positivity towards TADF emission. Furthermore, **BSBS-N1** showed a very high RISC of $1.9 \times 10^6\,s^{-1}$, which is much higher than all the v-DABNA analogs. OLEDs fabricated with the **BSBS-N1** showed a maximum EQE of 21% but fewer roll-off characteristics than other reported **MR-TADF** emitters. This design clearly shows that the heavy atoms can play a crucial role in increasing the RISC with proper molecular rigid structure.

Very recently, Colman et al. unveiled a new molecular design and developed **2B-DTACrs**[73]. The **2B-DTACrs** were synthesized using the commercially available starting materials in five steps. The designed **2B-DTACrs** resulted in a high synthetic yield of 58%. Due to the effect of *meta* B-π-B and O-π-O combination, the **2B-DTACrs** showed narrow FWHM of 21 nm in deep blue emission ($PL_{max}$ of 443 nm). The deep blue emission is expected as this contains the *meta* O-π-O than N-π-N (v-DABNA). Due to the rigid molecular design, the **2B-DTACrs** showed a very low $\Delta E_{ST}$ value of 0.16 eV, leading to a high RISC rate of $1.3 \times 10^5\,s^{-1}$. The OLED is fabricated with 5 wt% of **2B-DTACrs** in the **mCP** host. The device manifested a high EQE of 14.8% with $EL_{max}$ of 447 nm and FWHM of 26 nm. The corresponding CIE coordinates are ((0.150, 0.044), very close to BT2020 pure blue color requirements.

Precedently, Kido et al. reported pure blue emitters with the design considering the *meta* B-π-B and O-π-O and weak conjugation effect[74]. The designed emitter **m-DiNBO** was synthesized in three steps. The final step involves bromide-lithium

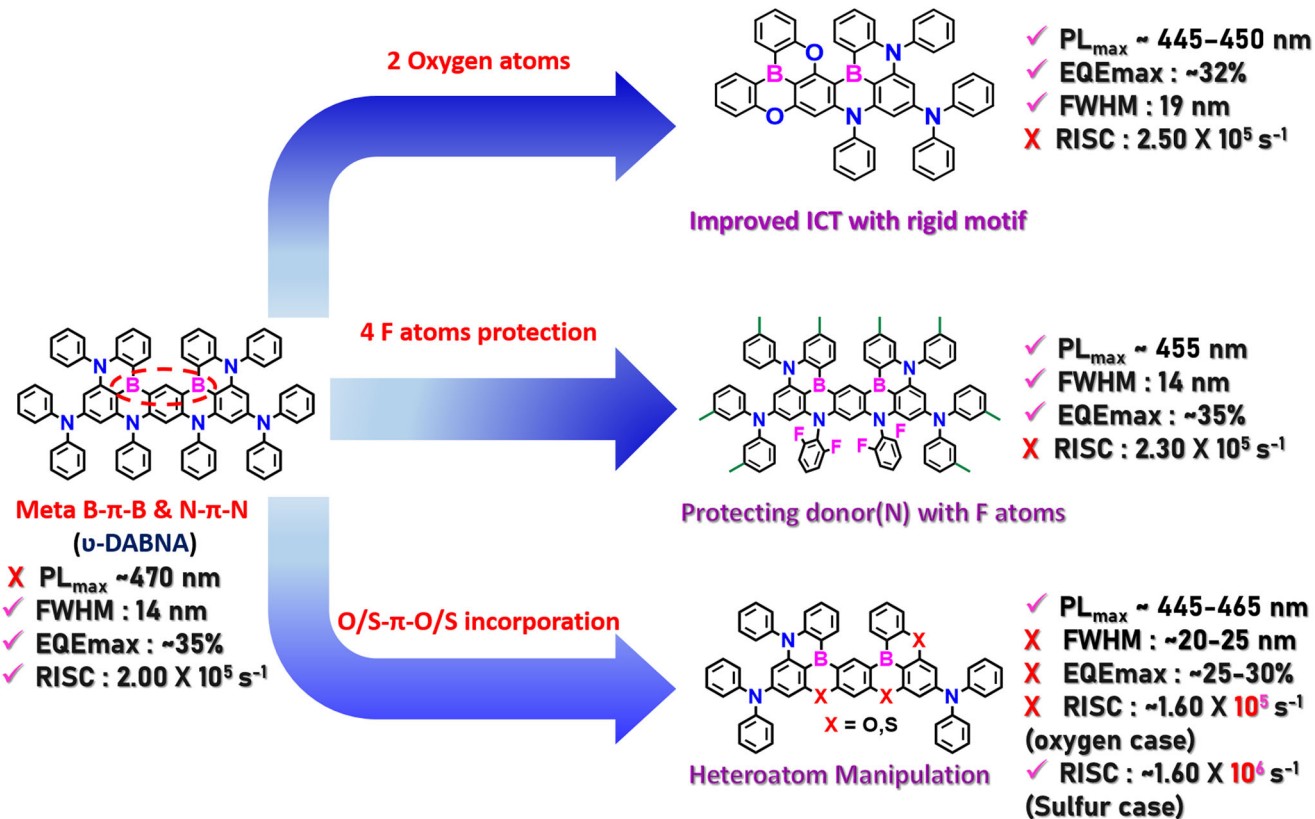

**Fig. 3 Pure blue DB-MR-TADF design approaches.** The design strategies for hypsochromic shift without compromising the color purity in **ν**-DABNA style DB-MR-TADF emitters.

exchange reaction-based one-pot borylation. The **m-DiNBO** manifested PL emission at 456 nm with FWHM of 17 nm and a high PLQY of 87%. When the **m-DiNBO** is deposited in the doped **mCBP** film, the PLQY is improved to 94%, indicating that small host-dopant interaction might be there. Finally, the OLEDs were fabricated with 3 wt% **m-DiNBO** -doped **mCBP** and showed a deep blue emission at 466 nm with an FWHM of 21 nm. The corresponding CIE coordinate of **m-DiNBO** is 0.098, with the EQE$_{max}$ of 24.2%. Mainly the **m-DiNBO** showed reduced efficiency roll-off characteristic compared to single boron **NBO** due to the lower delayed exciton lifetime ($\tau_d$) value of 31.4 μs for **m-DiNBO** than that for **NBO** ($\tau_d = 86.6$ μs). This design is very insightful as they used tert-butyl carbazole as their starting materials and created the **ν**-DABNA style type.

### Green to yellow DB-MR-TADF emitters

In 2020, Hatakeyama et al. unveiled a new molecular design by fusing the two MR analogs to get the double boron-based green emitter. The design emitter (**OAB-ABP-1**) represents a fully fused, resonating, extended π-skeleton by amalgamating the **ADBNA**[59] and **DOBNA**[75] substructures[76]. The **OAB-ABP-1** was synthesized like other **DB-MR-TADF** emitters, but the final step succeeded by using the BI$_3$ at 150 °C. As a result, the **OAB-ABP-1** gave an 18% synthetic yield by forming six C-B bonds. In addition, the ease of cyclization is favored due to mesitylene and methyl groups. As a result, the **OAB-ABP-1** shows pure green emission with PL$_{max}$ of 506 nm and the FWHM of 34 nm in a 1 wt% **PMMA** matrix. Notably, the film offers a high PLQY of 90%, which is far better than typical green D-A type emitters. Precedently, the $\Delta E_{ST}$ was found to be 0.12 eV, smaller than those of **DOBNA** and **ADBNA-Me-Mes** (0.18 and 0.20 eV, respectively). Due to the support of small $\Delta E_{ST}$, the **OAB-ABP-1** showed a

better RISC value than single boron-based **DOBNA** and **ADBNA-Me-Mes**[59]. Further, the solution-processed OLEDs were fabricated with the help of polymers. As a result, the device showed pure narrowband green emission with EL$_{max}$ of 505 nm with an FWHM of 33 nm. Consequently, the device revealed high EQE of 21.8% and 19.6% at 10 and 100 cd m$^{-2}$, respectively. The important central insight in this device is there is significantly less roll-off characteristic (<2.5%) observed, indicating their stability. Furthermore, the OLED showed a half-lifetime of 11 h at 300 nits. This was the first reported double boron-embedded green **MR-TADF** emitter. However, although this material showed outstanding performance, the FWHM is very broad, affecting the OLED's color purity.

Zheng et al. reported the two emitters (**DBON** and **DBSN**) by considering the para B-π-B and the introduction of hetero atoms (S and O) in para each other (Fig. 4)[77]. Due to the para B-π-B, the red-shifted emission can be expected compared to meta B-π-B (**ν**-DABNA). Furthermore, the hetero atoms' involvement can even improve the RISC process for efficient TADF. Both the emitters were prepared via successive simple nucleophilic reactions and one-pot lithiation-borylation-annulation. The final double boron cyclization has resulted in low yield, but the thermal stability is much higher as they have fused rigid molecular structures like **ν**-DABNA. Consequently, the **DBON/DBSN** exhibited PL$_{max}$ of 505/553 nm in the toluene solution with the FWHM of 20/28 nm. Compared to **OAB-ABP-1**, the **DBON** is better in the FWHM, which allows for high color purity in real applications. Notably, both the emitters showed a very small $\Delta E_{ST}$ of 0.13 eV, indicating their TADF phenomenon. Furthermore, due to the fused π core, both the emitters showed a high PLQY of 98%, indicating the limitation of the molecular nonradiative transitions. Due to the heavy atom sulfur, the **DBSN** showed yellow emission and a high RISC rate of $1.9 \times 10^5$ s$^{-1}$. This RISC rate is better compared to

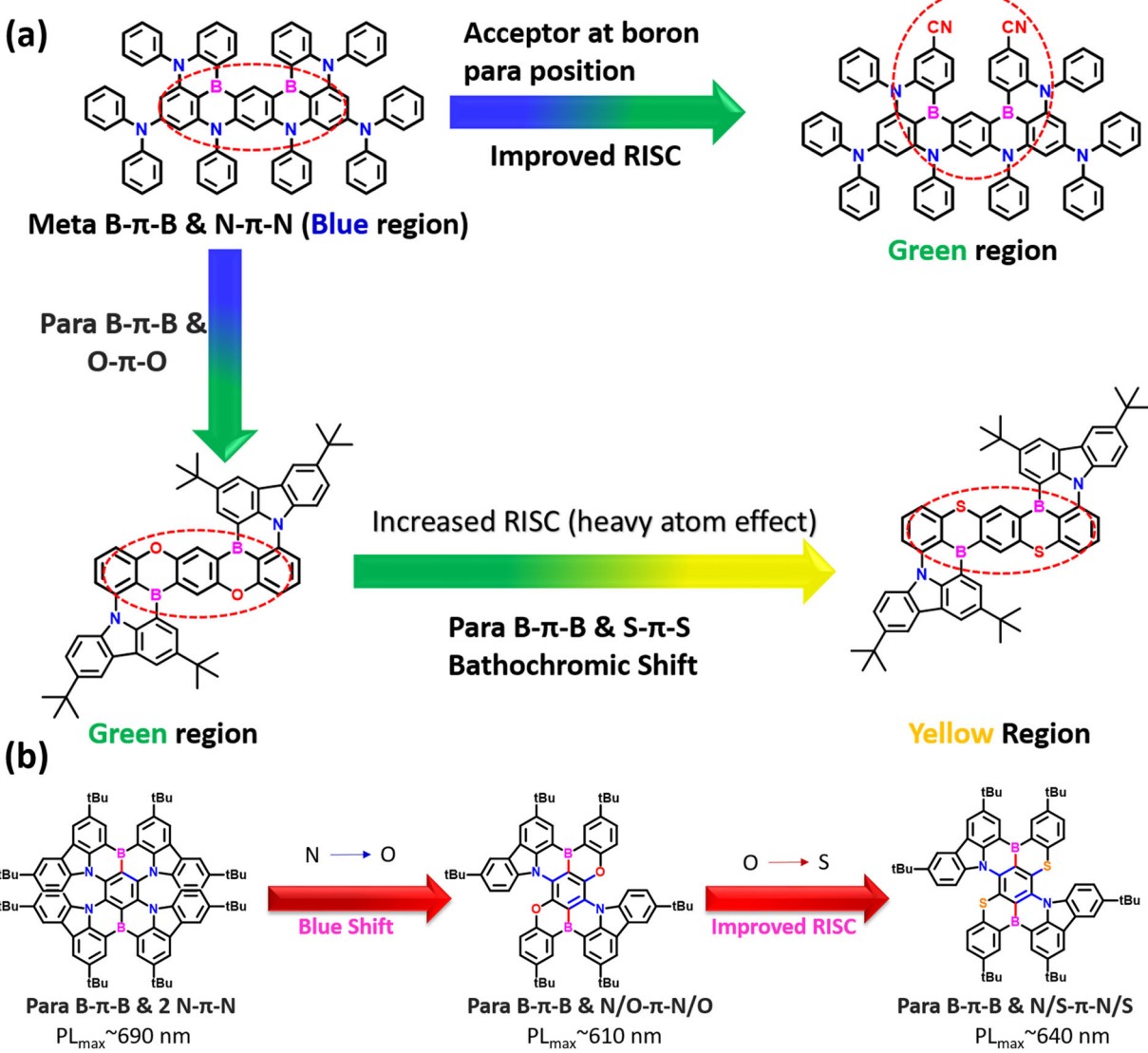

**Fig. 4 Pure green to red color DB-MR-TADF strategies.** Design approach for developing efficient (**a**) green to (**b**) red color **DB-MR-TADF** emitters.

other reported single boron **MR-TADF** emitters. Further, the OLEDs were fabricated using **mCBP** as host and **DBON** and **DBSN** as dopant. The device manifested high EQE of 26.7 and 21.8% for **DBON** and **DBSN**, respectively. The **DBON** and **DBSN** revealed pure green and yellow emissions with $EL_{max}$ of 510 and 556 nm and FWHM of 29 and 43 nm. Consequently, the corresponding CIE coordinates are (0.17, 0.68) and (0.42, 0.57) for **DBON** and **DBSN**, which match NTSE requirements. Due to the heavy atom (high RISC), the **DBSN** showed suppressed roll-off characteristics to **DBON**. Meanwhile, the same **DBON** is reported by Yue Wang et al., where named the material **DBNO**[78]. **DBNO** as a dopant in hyperfluorescence OLED presented tremendous EQE of 37.1% with the pure green emission at $EL_{max}$ of 504 nm and FWHM of 27 nm. Kido et al. recently synthesized **p-DiNBO** (same as **DBNO**) compared to **m-DiNBO** (meta B-π-B)[74]. The design allows for strong π-conjugation so that bathochromic shift can be retained. Compared to **m-DiNBO**, the **p-DiNBO** is ~45 nm red-shifted, indicating the strong effect of extended π-conjugation. The OLED architecture is utilized the same as the **m-DiNBO** device with 3 wt% in the **mCBP** host. The OLED revealed a high EQE of 21.6% with $EL_{max}$ of 513 nm and a CIE coordinate of (0.258, 0.665). However, compared to a single boron-based emitter (**NBO**), the **p-DiNBO** showed better EL performances.

Hatakeyama et al. recently unveiled a new molecular design for extremely narrowband green emission (**v-DABNA-CN-Me**)[57]. The design is based on incorporating cyano substitutions on a para to Boron atom in the previously reported blue narrow band emitter **v-DABNA**, allowing for bathochromic shifted emission. As the electron-withdrawing group, CN (accepting strength) at para positions to boron can decrease the LUMO energy level and increase the band gap. Initially, the two boron atoms were successfully embedded with the support of $BI_3$ at a low temperature (70 °C). Next, the palladium-catalyzed double cyanation was done to get the final emitter **v-DABNA-CN-Me** with a high synthetic yield of 43%. As a result, the **v-DABNA-CN-Me** manifested $PL_{max}$ of 496 nm with the FWHM of 17 nm only. This was the first material with such extremely narrow FWHM in the green region in the state-of-art **DB-MR-TADF** emitters. Due to the considerable $\Delta E_{ST}$ of 100 eV, the RISC was high as $1.0 \times 10^5 \text{ s}^{-1}$. The OLED is fabricated with the ET-type host material (99.5 wt% **DOBNA-Ph**) with 0.5 wt% of **v-DABNA-CN-Me**. The fabricated OLED showed pure narrowband green emission with $EL_{max}$ of 504 nm and FWHM of 23 nm. Notably, the device manifested a high EQE value of 31.9% at 10 cd/m². Interestingly, the roll-off characteristics are best among the green **MR-TADF** emitters as they possess the EQE values of 31.5% and 28.5%, even at 100 and

1000 cd/m$^2$, respectively. Consequently, the OLED showed a long lifetime (LT$_{80}$) of 59 h, which is better than previous **v**-DABNA-based designs. From this design, it is understood that the role-play of LUMO distribution can impact the bathochromic shift without losing color purity (Fig. 4).

### Red DB-MR-TADF emitters

For the first time, Yasuda et al. developed a pure red color emitter, namely, **BBCz-R**[79]. The design is based on enhancing the donating/accepting strengths by considering para B-π-B and N-π-N in one central benzene. The **BBCz-R** was synthesized in three steps, including the nucleophilic aromatic substitution followed by one-shot tandem borylation with BBr$_3$ and Hunig's base support. As a result, the **BBCz-R** resulted in a low synthetic yield of just 5% in the final boron-cyclization step. Notably, the 5% weight-loss decomposition temperatures ($T_d$) of **BBCz-R** are high as 533 °C, indicating good thermal stability. Furthermore, due to the 17-ring fused system, the **BBCz-R** showed a helical-type twisted configuration. Interestingly, the **BBCz-R** revealed narrowband pure red emission with the PL$_{max}$ of 615 nm and FWHM of 21 nm in the toluene solution. Further, the OLED is fabricated with the emissive layer of the **mCBP** host and the **BBCz-R** dopant (2 wt%). As a result, the OLED showed an EL$_{max}$ of 616 nm with an FWHM of 26 nm. Such narrow FWHM is attributed to the high molecular rigidity and increased fused ring systems. Notably, the OLED revealed a high EQE of 22% with the pure red color (CIE coordinates: (0.67, 0.33)). This is good work for the future development of red **MR-TADF** emitters. However, from the synthetic point of view, it is hard to achieve a high yield due to double borylation.

By considering a similar design strategy, Duan et al. developed two deep red **MR-TADF** emitters, **R-BN** and **R-TBN**[80]. The multiple B and N atoms were embedded in the central phenyl ring with the combination of B-phenyl-B and N-phenyl-N atoms, which enables the strong π-conjugation and allows for a narrow bandgap. The **R-BN** contains four carbazole units, whereas **R-TBN** has four tert-butyl substituted carbazole units. **R-BN** and **R-TBN** were synthesized in two steps from commercially available starting materials. The key boron cyclization was done using BBr$_3$ and Hunig base and achieved a high reasonable yield of ~40%. The synthesized emitters showed pure red emission with PL$_{max}$ of 662 nm (**R-BN**) and 692 nm (**R-TBN**) with FWHM of 38 nm in the toluene solution. Consequently, both emitters showed high PLQY of unity, with small $\Delta E_{ST}$ values of 0.18 eV for **R-BN** and 0.16 eV for **R-TBN**. By considering the long-delayed decay lifetimes of both emitters, the OLEDs were fabricated by adopting the emissive layer of the ternary system. The EML contained **CBP**: 30 wt% **Ir(mphmq)$_2$ tmd**: 3 wt% **R-BN/R-TBN**, where **CBP** as host and **Ir(mphmq)$_2$ tmd** were used as the phosphorescence sensitizer to assist the exciton in recycling under electrical excitation. Consequently, the OLED showed EL$_{max}$ of 664 nm (**R-BN**) and 686 nm (**R-TBN**) with small FWHMs of 48 and 49 nm, respectively. The corresponding CIE coordinates of (0.719, 0.280) and (0.721, 0.278) were observed for **R-BN** and **R-TBN**, respectively, indicating their potentiality as deep red emitters. Notably, both the emitters showed maximum EQE of 28.1 and 27.6% for **R-BN** and **R-TBN**, respectively. The OLED lifetime (LT$_{90}$) was manifested at 125 h and 151 h, respectively, under an initial luminance of 2000 cdm$^{-2}$. Even though these emitters showed good EQE and reasonable lifetime, their emission falls under the deep red region, which deviates from the pure red color requirements given by BT2020 and NTSE standards.

Very recently, Chuluo yang et al. developed a new molecular design strategy by using the weak donating ability of oxygen atoms at para each other to blue shift the prior **R-BN** designs (Fig. 5). So by considering the oxygen replacement effect, the para-boron/nitrogen fusing, the designed emitters (**BNO1**-**BNO3**) can fall in the pure red color region[58]. Furthermore, this design concept improves ICT characteristics by possessing para N-π-N, O-π-O, and B-π-B pairs. All three emitters were synthesized in three steps. The first two steps involve nucleophilic aromatic substitutions using carbazole and phenolic derivatives. The third and final step involves the n-butyl lithium-mediated borylation reaction using the BBr$_3$ and Hunig base. Finally, all three emitters gave a reasonable synthetic yield of over 30%, indicating that these materials can be easily scalable for commercial applications. Furthermore, these materials have high decomposition temperatures ($T_d$, corresponding to 5% weight loss) over 500 °C, indicating their tremendous thermal stability. All three emitters showed the PL$_{max}$ of 605–616 nm with the FWHM below 33 nm in toluene solution. The emission red-shifted from **BNO1** to **BNO3** due to the extensions of indeno and phenoxy peripheries. Notably, moderate $\Delta E_{ST}$s of 0.25–0.27 eV were determined based on their fluorescence and phosphorescent spectra. Precedently, all three emitters are highly anisotropic due to their quasi-planar molecular shape. This led to getting a high horizontal molecular ratio of ~86%. Consequently, all three emitters showed high PLQY of over 95% due to their strong MR effect with rigid molecular structure. Further, the phosphor sensitized OLEDs were fabricated with the EML of 1–3 wt% of emitters and 20 wt% of **PO-01** in the **DMIC-TRZ** host. Among all materials, the **BNO3** manifested tremendous performances as it exhibited EQE$_{max}$ of 36.1% and it maintained a very high EQE of 28.6% at 10000 cd/m$^2$. Importantly, all the materials exhibited pure-red emission with CIE x ~0.64–0.67. Further, **BNO1** showed decent operational stability with LT$_{90}$ of 49.8 h. This design concept clearly shows that incorporating heteroatoms in rigid molecular structures can lead to ultralow efficiency roll-off, ultrahigh brightness, high color purity, and good device lifetime.

To extend the red **DB-MR-TADF** emitters, recently, L. Wang et al. developed two emitters, namely, **DBNS** and **DBNS-tBu**[81]. The design is based on utilizing the sulfur atoms instead of oxygen atoms in the **BNO1** core. The synthesis followed the same **BNO1** emitter, but thiophenol was used instead of phenol groups. The final emitters have resulted in reasonable synthetic yields. It is noted that the sulfur atoms showed higher HOMO contribution than nitrogen atoms, so that RISC can be improved. The **DBNS** showed the PL$_{max}$ of 631 nm, whereas **DBNS-tBu** showed 641 nm. The 10 nm redshifted for **DBNS-tBu** is due to the incorporation of tert-butyl groups. Consequently, both the emitters exhibited narrow FWHM of ~40 nm with a high PLQY of ~85%. The FWHM is increased compared to oxygen-bridged type **BNO1** due to the twisted geometry of **DBNS** and a heavy sulfur atom. Precedently, the heavy atom helped to get the high RISC rate of ~2.2 × 10$^5$ s$^{-1}$ for both the emitters. Even though the emitters revealed twisted geometries, the trigonal planarity of the overall core leads to aggregation-induced emission in **DBNS**. In contrast, when tert-butyl groups are incorporated, AIE slightly alleviates and maintains the narrow FWHM. The solution-processed OLED was fabricated by adopting the EML with a ternary system of **H2**: 10 wt% **R-D2**: 1–3 wt% **DBNS** or **DBNS-tBu**. Among both emitters, the **DBNS-tBu** revealed a high EQE of 7.8% with an EL$_{max}$ of 616 nm. The EL emission is blue-shifted compared to PL emission due to higher S$_1$ energy levels in the **H2** host matrix[82] than in dichloromethane solution. However, from this design, it is understood that the heavy atoms, especially sulfur, allow increasing the TADF mechanism with the high RISC rate constant values. The detailed photophysical and OLED performances were depicted in Tables 1 and 2.

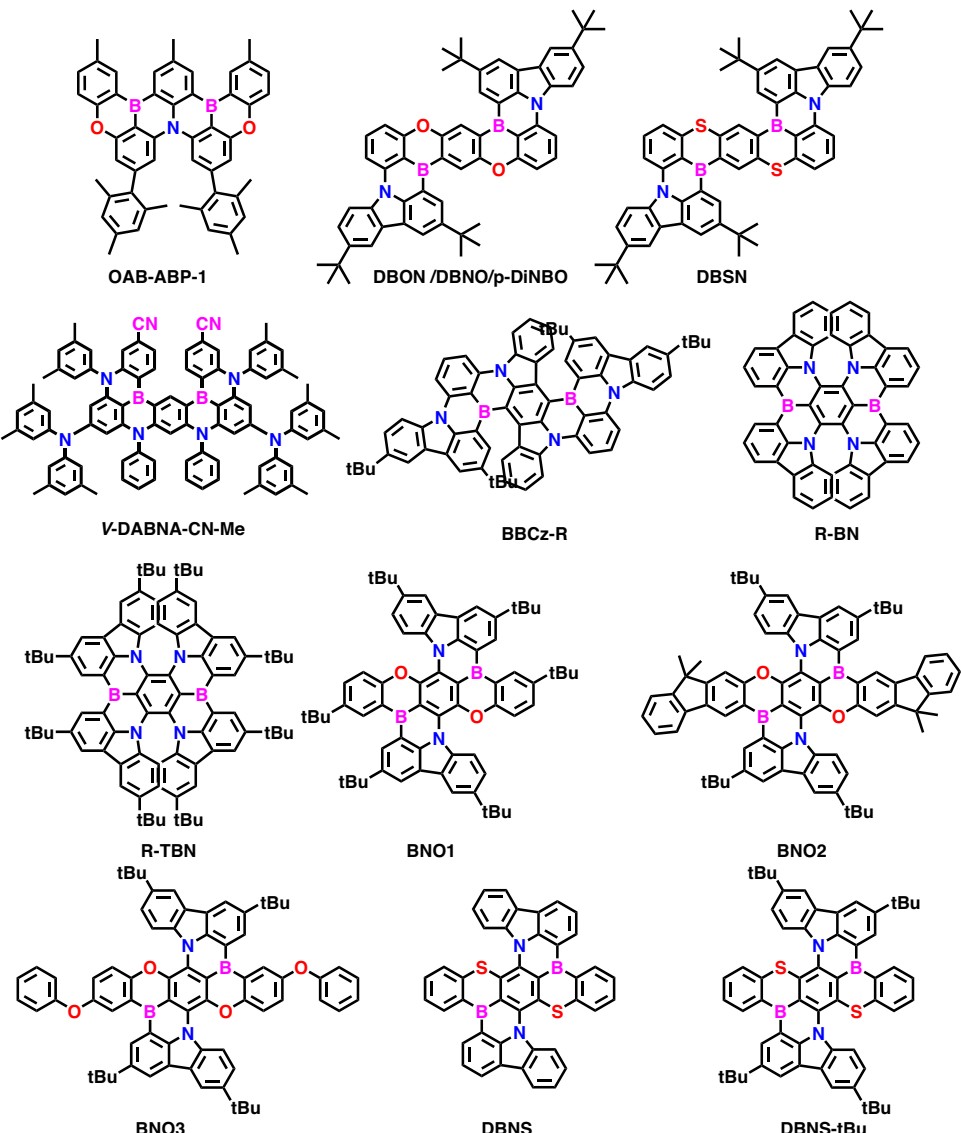

**Fig. 5 Chemical structures of green to deep red MR-TADF.** Molecular structures of the green to deep red **DB-MR-TADF** emitters.

## Overall synthetic aspects for borylation

There are two types mainly utilized for the synthesis of the **DB-MR-TADF** emitters. These are (i) *BuLi*-based tandem borylation using Hunig base; (ii) One shot double borylation by using BBr$_3$/BI$_3$ and reducing aromatic solvents[49,83–90]. In type-1, under the inert atmosphere in the reducing solvents, the halogen will be replaced by using lithiation, and then transmetalation was done by using BBr$_3$ with the temperature controls[90–93]. And then finally, the tandem bora Friedel-craft reactions using the Hunig base. Whereas in the case of Type-2, the boron cyclization was done by one shot using the BBr$_3$/BI$_3$ with the support of aromatic solvents like toluene, chlorobenzene, o-dichlorobenzene, and 1,2,4-trichlorobenzene[93–96]. Both types allow getting the **C-B** formations. The conceptual insights of both synthetic aspects are provided in Fig. 6.

## Summary and outlook

Neoteric developments strongly reveal that constructing high-performance OLEDs with high energy conversion efficiency comparable to phosphorescent (metal-based) counterparts by using purely organic semiconductors is feasible. A variety of strictly organic emitters based on enormous building blocks has already been confirmed with the capacity to make highly efficient OLEDs. Among them, the TADF materials were efficient as they could achieve 100% IQE. However, due to the vibrational spectrum broadening, a new era of **MR-TADF** materials is blooming in the current OLED field research. This review discusses the effect of two boron atoms and the summarized progress of double boron-embedded multi-resonant TADF emitters based in all color regions. We mainly focus on how to design the materials with valuable conceptual insights and good synthetic processes.

In the case of blue color, the **ν-DABNA** type emitters showed tremendous performances as it involves the meta B-π-B/N-π-N combinations at a central electron-rich phenyl ring. The performances such as narrow FWHM (~15 nm), high PLQY, and high RISC have succeeded already. The color point of view in the blue region is resolved already with proper design concepts and high EL performances as they also show EQE$_{max}$ of ~35%. However, the lifetime for blue materials has still lacking in consideration for future developments. Meanwhile, in the case of the green region, the **ν-DABNA-CN-Me** showed outstanding performance as it showed good color purity (FWHM 23 nm) compared to previous designs. The alternative way to design the efficient green **DB-MR-**

**Table 1 Detailed photo physical properties of DB-MR-TADF emitters.**

| Emitter | PLmax (nm) | FWHM (nm) | PLQY (%) | HOMO (eV) | LUMO (eV) | $S_1$ (eV) | $T_1$ (eV) | $\Delta E_{ST}$ (eV) | $\tau_p$ (ns) | $\tau_d$ (μs) | $k_{RISC}$ ($10^5$ s$^{-1}$) | Ref. |
|---|---|---|---|---|---|---|---|---|---|---|---|---|
| ADBNA-Me-Mes | 482[b] | 33[b] | 89 | - | - | 2.59 | 2.39 | 0.20 | 6.9 | 165 | 0.07 | 59 |
| ADBNA-Me-Tip | 479[b] | 34[b] | 88 | - | - | 2.60 | 2.41 | 0.19 | 6.8 | 147 | 0.09 | 60 |
| 3 | 482[e]/485[a] | - | 71 | -5.62 | -2.55 | 2.56 | 2.38 | 0.18 | - | - | - | |
| 4a | 485[e]/488[a] | - | 88 | -5.69 | -2.69 | 2.54 | 2.35 | 0.19 | - | - | - | |
| 4b | 487[e]/491[a] | - | 85 | -5.79 | -2.76 | 2.53 | 2.34 | 0.19 | - | - | - | |
| 5a | 483[e]/485[a] | - | 82 | -5.61 | -2.64 | 2.56 | 2.38 | 0.18 | - | - | - | |
| 5b | 486[e]/487[a] | - | 81 | -5.61 | -2.71 | 2.55 | 2.38 | 0.17 | - | - | - | |
| 6a | 487 | - | 19 | -5.05 | -2.54 | 2.55 | 2.38 | 0.17 | - | - | - | |
| 6b | 495 | - | 18 | -5.06 | -2.53 | 2.51 | 2.38 | 0.13 | - | - | - | |
| B2 | 455[a] | 32[a] | 53 | - | - | 2.73 | 2.55 | 0.18 | - | - | - | 61 |
| ν-DABNA | 468[c] | 14[c] | 90[d] | - | - | 2.64 | 2.62 | 0.02 | 41 | 4.3 | 2.00 | 56 |
| ν-DABNA-O-Me | 464[a] | 24[a] | 90 | - | - | 2.75 | 2.72 | 0.03 | 5.1 | 7.7 | 1.60 | 62 |
| BOBO-Z | 441[c]/445[d] | 15[c]/18[d] | 76 | - | - | - | - | 0.10 | 2.3 | 7.7 | 0.70 | 63 |
| BOBS-Z | 453[c]/457[d] | 21[c]/24[d] | 94 | - | - | - | - | 0.12 | 1.1 | 7.6 | 8.60 | |
| BSBS-Z | 460[c]/464[d] | 20[c]/22[d] | 93 | - | - | - | - | 0.12 | 1.0 | 6.7 | 16.00 | |
| m-ν-DABNA | 464[c] | 14[c] | 90 | -5.55 | -2.93 | 2.81 | 2.74 | 0.07 | 8.43 | 3.09 | 2.30 | 64 |
| 4F-ν-DABNA | 457[c] | 14[c] | 90 | -5.66 | -3.03 | 2.84 | 2.79 | 0.05 | 8.72 | 3.12 | 2.28 | |
| 4F-m-ν-DABNA | 455[c] | 14[c] | 89 | -5.61 | -2.96 | 2.86 | 2.79 | 0.07 | 8.20 | 3.19 | 2.10 | |
| TPD4PA | 445[c] | 19[c] | 88[d] | -5.54 | -2.79 | 2.93 | 2.88 | 0.05 | 7.82 | 4.69 | 2.51 | 65 |
| tBu-TPD4PA | 451[c] | 19[c] | 90[d] | -5.52 | -2.81 | 2.90 | 2.84 | 0.06 | 8.07 | 5.55 | 2.44 | |
| BN3 | 456[c] | 17[c] | 98[d] | - | - | 2.79 | 2.64 | 0.15 | 1.30 | 17.8 | 2.55 | 71 |
| BSBS-N1 | 473[c]/478[f] | 21[c]/24[f] | 59[c]/89[f] | - | - | 2.62 | 2.49 | 0.13 | 0.90 | 5.60 | 19.00 | 72 |
| 2B-DTAcrs | 443[c]/448[f] | 21[c]/24[f] | 74 | -5.51 | -2.46 | - | - | 0.16 | 2.0 | 13.1 | 1.30 | 73 |
| m-DiNBO | 456[c] | 17[c] | 94 | -5.30 | -2.60 | 2.75 | 2.69 | 0.06 | 3.4 | 31.4 | 0.31 | 74 |
| OAB-ABP-1 | 506[a] | 34[a] | 90 | - | - | - | - | 0.12 | 9.9 | 32 | 0.40 | 76 |
| DBNO/DBON | 505[c] | 20[c] | 98 | -5.40 | -2.98 | 2.46 | 2.33 | 0.13 | 3.7 | 24.6 | 0.80 | 77 |
| ν-DABNA-CN-Me | 496[c] | 17[c] | 86 | - | - | 2.56 | 2.50 | 0.06 | 4.9 | 10.0 | 1.00 | 57 |
| DBSN | 553[c] | 28[c] | 98 | -5.28 | -3.05 | 2.24 | 2.11 | 0.13 | 4.3 | 25.7 | 1.90 | 77 |
| BBCz-R | 619[d] | 27[d] | 79 | - | - | 2.09 | 1.90 | 0.19 | 5.1 | 53 | 0.12 | 79 |
| R-BN | 662[c] | 38[c] | 100 | -4.76 | -3.06 | 1.87 | 1.69 | 0.18 | 12.0 | 16.6 | 0.67 | 80 |
| R-TBN | 692[c] | 38[c] | 100 | -4.69 | -3.00 | 1.79 | 1.63 | 0.16 | 14.2 | 46.4 | 0.25 | |
| BNO1 | 605[c]/610[g] | 32[c]/35[g] | 96 | -4.86 | -2.83 | 2.15 | 1.90 | 0.25 | - | - | - | 58 |
| BNO2 | 609[c]/618[g] | 32[c]/37[g] | 95 | -4.85 | -2.85 | 2.13 | 1.86 | 0.27 | - | - | - | |
| BNO3 | 616[c]/624[g] | 33[c]/38[g] | 96 | -4.84 | -2.85 | 2.11 | 1.85 | 0.26 | - | - | - | |
| DBNS | 631[c] | 40 | 80 | - | - | 2.05 | 1.85 | 0.20 | 14.4 | 11.2 | 2.10 | 81 |
| DBNS-tBu | 641[c] | 39 | 85 | - | - | 2.02 | 1.83 | 0.19 | 19.1 | 10.2 | 2.20 | |

Blue to Deep red color-based double boron embedded MR-TADF emitters.
[a]PMMA film.
[b]DOBNA-OAr film.
[c]Toluene.
[d]Film.
[e]CH$_2$Cl$_2$.
[f]mCBP film.
[g]1 wt% doped in DMIC-TRZ film.

**Table 2 Detailed device characteristics of DB-MR-TADF emitters.**

| Emitter | HOST | EQE$_{max}$ (%) | EQE$_{100}$/ EQE$_{1000}$ (%) | CE (cd/A) | PE (lm/W) | EL/FWHM (nm) | CIE (x, y) | Ref. |
|---|---|---|---|---|---|---|---|---|
| ADBNA-Me-Mes | DOBNA-OAr | 16.2 | 11.1/- | 24.5 | 19.3 | 481/32 | 0.10,0.27 | [59] |
| ADBNA-Me-Tip | DOBNA-OAr | 21.4 | 15.4/- | 34.7 | 28.7 | 480/33 | 0.11,0.29 | |
| B2 | mCBP | 18.3 | 12.6/- | 16.7 | 13.8 | 460/37 | 0.13,0.11 | [61] |
| ν-DABNA | DOBNA-OAr | 34.4 | 32.8/26.0 | 31.0 | 25.6 | 469/18 | 0.12,0.11 | [56] |
| ν-DABNA-O-Me | DOBNA-Tol | 29.5 | 28.8/26.9 | 24.6 | 22.7 | 465/23 | 0.13,0.10 | [62] |
| BOBO-Z | mCBP | 13.6/16.6[a] | 9.8/3.3 | 7.2/8.3[a] | 5.0/5.3[a] | 445/18 | 0.15,0.04 | [63] |
| BOBS-Z | mCBP | 26.9/33.1[a] | 24.0/15.0 | 16.7/19.9[a] | 12.9/15.4[a] | 456/23 | 0.14,0.06 | |
| BSBS-Z | mCBP | 26.8/32.2[a] | 24.0/15.9 | 23.2/24.8[a] | 15.0/18.4[a] | 463/22 | 0.13,0.08 | |
| m-ν-DABNA | DBFPO | 36.2 | 20.4[c] | 32.1 | – | 471/18 | 0.12,0.12 | [64] |
| 4F-ν-DABNA | DBFPO | 35.8 | 16.7[c] | 26.8 | – | 464/18 | 0.13,0.08 | |
| 4F-m-ν-DABNA | DBFPO | 33.7 | 17.9[c] | 24.9 | – | 461/18 | 0.13,0.06 | |
| TPD4PA | mCBP-CN | 30.7 | 30.6/17.8 | 15.7 | – | 455/29 | 0.14,0.06 | [65] |
| tBu-TPD4PA | mCBP-CN | 32.5 | 30.9/20.5 | 19.4 | – | 460/29 | 0.14,0.07 | |
| BN3 | DBFPO:25 wt% 3Cz2BN | 37.6 | 34.0/26.2 | 27.5 | 19.0 | 458/23 | 0.14,0.08 | [71] |
| BSBS-N1 | mCBP | 21.0 | -/- | – | – | 478/25 | 0.11,0.22 | [72] |
| 2B-DTAcrs | mCP | 14.8 | -/- | – | – | 447/26 | 0.15,0.04 | [73] |
| m-DiNBO | mCBP | 24.2 | -/- | – | – | 466/21 | 0.12,0.10 | [74] |
| OAB-ABP-1 | Polymer B | 21.8 | 19.6/17.4 | 53.3 | 45.3 | 505/33 | 0.12,0.63 | [76] |
| DBON | mCBP | 26.7 | 20.2/12.0 | 94.1 | 75.8 | 510/29 | 0.17,0.68 | [77] |
| ν-DABNA-CN-Me | DOBNA-Ph | 31.9 | 31.5/28.5 | 88.9 | 93.6 | 504/23 | 0.13,0.65 | [57] |
| DBSN | mCBP | 21.8 | 20.6/16.9 | 84.7 | 71.9 | 556/43 | 0.42,0.57 | [77] |
| BBCz-R | mCBP | 22.0 | -/- | – | – | 616/26 | – | [79] |
| R-BN | CBP: 30 wt% Ir(mphmq)2 tmd | 28.1 | -/- | – | – | 664/48 | 0.72,0.28 | [80] |
| R-TBN | CBP: 30 wt% Ir(mphmq)2 tmd | 27.6 | -/- | – | – | 686/49 | 0.72,0.28 | |
| BNO1 | DMIC-TRZ | 35.6 | -/31.1 | 59.4 | 66.7 | 610/39 | 0.64,0.34 | [58] |
| BNO2 | DMIC-TRZ | 34.4 | -/29.8 | 46.5 | 52.2 | 618/39 | 0.65,0.35 | |
| BNO3 | DMIC-TRZ | 36.1 | -/32.1 | 43.4 | 48.7 | 625/40 | 0.66,0.34 | |
| DBNS[b] | H2 | 5.8 | -/4.4 | 7.2 | – | 613/66 | 0.64,0.34 | [81] |
| DBNS-tBu[b] | H2 | 7.8 | -/5.8 | 9.2 | – | 616/65 | 0.65,0.35 | |

Device performances of pure blue to deep red color-based DB-MR-TADF emitters.
[a]Device equipped with a microlens array film.
[b]Solution processed device.
[c]EQE400 (%).

**Fig. 6 Possible synthetic routes of double borylations.** The synthetic pathways utilized for most of the **DB-MR-TADF** emitters.

**TADF** is by combining para B-π-B and O-π-O. The **ν-DABNA-CN-Me** utilized the same meta combination of B/N only like blue emitter **ν-DABNA**. Still, the CN group incorporated para position to boron, which affected the LUMO and showed green emission with good RISC improvements. However, green **MR-TADF** emitters require still more development focusing on pure color and even good EL characteristics. To design the red **DB-**

**MR-TADF** emitters, the central phenyl ring should be surrounded by para B-π-B, N-π-N, and O-π-O each other. The emitter **BNO1** recently showed good red emission performances. However, the FWHM can be reduced (below 30 nm) in the coming days by considering the other design concepts as well. Overall, even from a synthetic point of view in **DB-MR-TADF** designs, both halogen included /halogen-free already have proper

design analysis for developing the efficient and high color purity **DB-MR-TADF** emitters in all regions. So maybe in future days, utilizing these **DB-MR-TADF** emitters, highly efficient stable devices can be developed with proper device concepts. Hyper-fluorescence (HF) has been a keen technology in recent years, so by developing highly efficient stable emitters, the stable HF-OLED can be retained for future display applications, satisfying commercial TV standards. We believe this review article can guide the pathways for designing efficient and stable **MR-TADF** emitters with high color purity for futuristic developments in OLED materials research.

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

## Acknowledgements

This work was supported by the Technology Innovation Program (20006464, Development of TADF sensitized fluorescence RGB emitting layer materials with high color

purity and high efficiency for BT 2020) funded by the Ministry of Trade, Industry & Energy (MOTIE, Korea).

## Author contributions

K.R.N. conceptualized the paper and wrote the draft. H.I.Y. searched the literature and helped in drawing the figures. J.H.K. supervised the overall manuscript.

## Competing interests

The authors declare no competing interests.
