## [Peer Review File · Communications Chemistry]

REVIEWERS' COMMENTS:

Reviewer #1 (Remarks to the Author):

The manuscript reviews recently-developed MR-TADF, mainly focusing on double-boron (DB) types. Such a review is timely and I strongly recommend the publication after addressing the following points.

1. The authors focus on DB-type MR-TADF. To clarify the advantage, comparison tables of single boron (SB) and DB MR-TADF are helpful for readers. The data of DB-MR-TADFs are already provided in Tables 1 and 2, so please include such Tables for SB-MR-TADFs.
2. "A further interesting fact is that the TADF mechanism occurred via three steps, including (i) IC from T1 to T2, (ii) RISC from T2 to S1, and (iii) fluorescence from S1." (Lines 236-238): This is not a special case. Even for the original DABNA-1, RISC occurs via T2. Please check Commun. Chem., 5, 53, (2022); Doi: 10.1038/s42004-022-00668-6.
3. In the Tables, data on efficiency roll-offs are lacking.
4. V-DABNA and nu-DABNA (here, I cannot use Greek characters) indicate different molecules, but are mixed up in the manuscript. Please correct them.
5. Line 60: efficiency -> external quantum efficiency (EQE)
6. "the extremely small ΔE_{ST} of 100 eV" (Line 366): 100 eV is too large. Please correct it.
7. Figure 5: Methyl groups are lacking in v-DABNA-CN-Me. (This is also not V-DABNA-CN-Me).
8. There are many places where letters are capitalized, even though they are not at the beginning of sentences. Please also recheck the English as a whole.

Reviewer #2 (Remarks to the Author):

It was mainly reported from a synthetic point of view, and many double-boron MR-TADF materials have not been reported yet, so it was not covered in depth, but it was simply explained and organized in terms of synthesis and physical properties of double-boron MR TADF in RGB. However, In terms of design concept, I suggest that additional explanation is needed from the orbital point of view to see if there was any change in NTO or HONTO/LUNTO depending on the type of substituent.

Reviewer #3 (Remarks to the Author):

In this article, Kwon and coworkers reviews recent progress in the DB-MR-TADF emitters with particular attention to molecular design concepts, optoelectronic properties, synthetic routes, and OLED performances with future prospects for real-world applications. The summary is meaningful and comprehensive. This work can be published after revisions.

1. The publication date of Reference 74 is earlier than Reference 73, so Reference 74 should be first described in detail.
2. The relative references should be cited, including Adv. Mater. 2021, 33, 2100652; CCS Chem. 2022, 4, 2065–2079 etc.

Research Article No: COMMSCHEM-22-0354

Title: Double boron-embedded multiresonant thermally activated delayed fluorescent materials for organic light emitting diodes

“Point-to-Point” responses to reviewers’ comments

We are grateful for the careful evaluation and thoughtful comments from the reviewers & editorial board, which have helped to improve the manuscript. We have attached our point-to-point responses. The original comments from the referee are in black. The response to the comment is in **dark red**. The corresponding changes made in manuscript are in **blue**.

Comments & Responses (Reviewer #1):

The manuscript reviews recently-developed MR-TADF, mainly focusing on double-boron (DB) types. Such a review is timely and I strongly recommend the publication after addressing the following points.

We thank you for your careful reading and the evaluation to our manuscript.

1. The authors focus on DB-type MR-TADF. To clarify the advantage, comparison tables of single boron (SB) and DB MR-TADF are helpful for readers. The data of DB-MR-TADFs are already provided in Tables 1 and 2, so please include such Tables for SB-MR-TADFs.

Response: Thank you for your suggestion. Actually in this review, we mainly focused on the DB-MR-TADF emitters instead of SB-MR-TADF emitters. In introduction, we mentioned that the DB-MR-TADF emitters are having many advantages such as narrow FWHM, efficient TADF characteristics and extremely small ΔE_{ST} and of course high OLED performances compare to SB-MR-TADF with relevant reference were provided. So we thought that the flow of the manuscript were sufficient by focus on perspective way for real molecular designs. So the Tables on SB-MR-TADF were not much important, as we have cited most of the SB-MR-TADF based emitters for OLEDs by comparing the DB-MR-TADF as well.

2. "A further interesting fact is that the TADF mechanism occurred via three steps, including (i) IC from T1 to T2, (ii) RISC from T2 to S1, and (iii) fluorescence from S1." (Lines 236-238): This

is not a special case. Even for the original DABNA-1, RISC occurs via T2. Please check Commun. Chem., 5, 53, (2022); Doi: 10.1038/s42004-022-00668-6.

Response: As reviewer mentioned, the conventional DABNA-1 emitter also manifest the RISC via T2. That also true. And we did not mentioned that in DB-MR-TADF it is special case. We said that in that particular work (which is TPD4PA and tBu-TPD4PA emitters), the RISC mechanism were analyzed through DFT like same work (Commun. Chem., 5, 53, (2022)). We have cited this work already in our manuscript as well.

3. In the Tables, data on efficiency roll-offs are lacking.

Response: As the MR-TADF emitter's shows severe efficient roll-off characteristics compare to conventional D-A type TADF emitters, the work is still undergoing for the explorations in this field. So most of the device showed severe roll-off characteristics as some OLEDs might be dead even at 1000cd/m². So we did not include the roll-off data in the particular table. Moreover we focus on colour point of view for designs based on DB-MR-TADF emitters.

4. V-DABNA and nu-DABNA (here, I cannot use Greek characters) indicate different molecules, but are mixed up in the manuscript. Please correct them.

Response: Thank you for your suggestion. We have revised the manuscript with Greek letter like ν -DABNA in the whole manuscript.

5. Line 60: efficiency -> external quantum efficiency (EQE)

Response: Thank you for your suggestion. We have the revised the EQE as reviewer suggested in updated manuscript.

6. "the extremely small Δ EST of 100 eV" (Line 366): 100 eV is too large. Please correct it.

Response: Thank you for your suggestion. We have corrected this sentence in revised manuscript.

7. Figure 5: Methyl groups are lacking in ν -DABNA-CN-Me. (This is also not V-DABNA-CN-Me).

Response: Thank you for your suggestion. We have updated the figure with methyl groups in the revised version.

8. There are many places where letters are capitalized, even though they are not at the beginning of sentences. Please also recheck the English as a whole.

Response: Thank you for your suggestion. We have crosschecked the whole manuscript and updated clearly all the English letter and words in the revised version.

Comments & Responses (Reviewer #2):

It was mainly reported from a synthetic point of view, and many double-boron MR-TADF materials have not been reported yet, so it was not covered in depth, but it was simply explained and organized in terms of synthesis and physical properties of double-boron MR TADF in RGB. However, In terms of design concept, I suggest that additional explanation is needed from the orbital point of view to see if there was any change in NTO or HONTO/LUNTO depending on the type of substituent.

Response: We thank you for your careful reading and the evaluation of our manuscript. As you can see from Figures 3 and 4, the design strategies are focused on the RGB color point of view. Mainly figure-3 explains the weak donating ability strategies towards hypsochromic shift without compromising the color purity in v-DABNA style DB-MR-TADF emitters as the v-DABNA exhibited efficient performances in DB-MR-TADF type emitters for OLEDs. Furthermore, we mentioned in detailed explanations in each work, mentioning the effect of either electron donating (nitrogen to oxygen atoms) or electron-withdrawing (Fluorine) in particular positions based on rigid Boron atoms and considering the HOMO/LUMO density lobes (DFT analysis) also towards pure color. Whereas similar explanations were given already in green (Like CN at boron para position) and red color (para Boron- π -B combination) based on NTO analysis (HOMO/LUMO effect) in particular each molecule explanation areas.

Comments & Responses (Reviewer #3):

In this article, Kwon and coworkers reviews recent progress in the DB-MR-TADF emitters with particular attention to molecular design concepts, optoelectronic properties, synthetic routes, and OLED performances with future prospects for real-world applications. The summary is meaningful and comprehensive. This work can be published after revisions.

We thank you for your careful reading and the evaluation to our manuscript.

1. The publication date of Reference 74 is earlier than Reference 73, so Reference 74 should be first described in detail.

Response: Thank you for your suggestion. However, in reference 74, initially, the DBON only reported. In reference 73, the DBON (same core as in ref.74) and DBSN were also reported. In our discussion, we compared both the oxygen and sulfur atom importance, so we chose the reference 73 initially to explain both (DBON and DBSN), and then the hyperfluorescence device for DBON would be suitable for improving the efficiency would be reasonable.

2. The relative references should be cited, including Adv. Mater. 2021, 33, 2100652; CCS Chem. 2022, 4, 2065–2079 etc.

Response: As the reviewer suggested we have cited the both papers while describing the importance of MR-TADF emitters in introduction part.